# When Mast Cells Run Amok: A Comprehensive Review and Case Study on Severe Neonatal Diffuse Cutaneous Mastocytosis

**DOI:** 10.3390/genes14112021

**Published:** 2023-10-29

**Authors:** Emilian-Gheorghe Olteanu, Mihaela Bataneant, Maria Puiu, Adela Chirita-Emandi

**Affiliations:** 1Research Center for Pharmaco-Toxicological Evaluations, Faculty of Pharmacy, “Victor Babes” University of Medicine and Pharmacy, Piaţa Eftimie Murgu Nr. 2, 300041 Timisoara, Romania; olteanu.gheorghe@umft.ro; 2Center for Research and Innovation in Personalized Medicine of Respiratory Diseases, “Victor Babes” University of Medicine and Pharmacy, 300041 Timisoara, Romania; 3Discipline of Pediatrics, “Victor Babes” University of Medicine and Pharmacy, 300041 Timisoara, Romania; 4Department of Oncology and Hematology, Clinical Emergency Hospital for Children “Louis Turcanu”, 300041 Timisoara, Romania; 5Discipline of Genetics and Center of Genomic Medicine, “Victor Babes” University of Medicine and Pharmacy, 300041 Timisoara, Romania; maria_puiu@umft.ro (M.P.); adela.chirita@umft.ro (A.C.-E.); 6Regional Center of Medical Genetics Timis, Clinical Emergency Hospital for Children “Louis Turcanu”, 300041 Timisoara, Romania

**Keywords:** neonatal diffuse cutaneous mastocytosis, *KIT* gene, case report, systemic mastocytosis, review

## Abstract

Neonatal diffuse cutaneous mastocytosis (NDCM) is defined as the infiltration of the epidermis by a clonal proliferation of mast cells, observed at birth, without initial signs of systemic involvement. The typical driver mutation is in the *KIT* gene. We report a rare case of a boy, born at term, already presenting at birth with generalized subcutaneous nodules on the face, scalp, trunk, back, hands, and feet. The spleen, liver, and inflammatory markers were normal at birth. Tryptase was significantly elevated. A bone marrow biopsy showed no mast cell involvement at age 2 months. A punch biopsy at age 2 months revealed CD117-positive cells diffusely infiltrating the skin, with subsequent DNA NGS sequencing for the formalin-fixed paraffin embedded tissue (FFPE) identifying the pathogenic NM_000222.3:c.1504_1509dup; p.(Ala502_Tyr503dup) variant in the *KIT* gene previously associated with cutaneous mastocytosis. At 2 years follow-up, he had splenomegaly and multiple cervical and inguinal adenopathy, while the skin nodules persisted, especially on the scalp with accompanying pruritus. He received oral and local sodium cromoglycate, oral antihistamines, antibiotic cream for skin infection, and iron supplementation; however, compliance to treatment was relatively low. The prognosis is difficult to predict, as he developed systemic involvement, failure to thrive, and mild psychomotor delay. A case aggregation of NDCM reported in the literature was performed to provide a comprehensive overview of this rare pathology, to better understand the prognosis. NDCM is a life-threatening disease with severe complications. Almost half had severe complications, such as mast hepatosplenomegaly, adenopathy, bacterial infections, mast cell leukaemia, and systemic involvement.

## 1. Introduction

Cutaneous mastocytosis has been classified into three main types: diffuse cutaneous mastocytosis (DCM), urticaria pigmentosa (UP), and solitary mastocytoma [1]. Of them, DCM is the rarest and the most severe form. Neonatal diffuse cutaneous mastocytosis (NDCM) is a very rare form of the disease, characterized by the infiltration of the epidermis by a clonal proliferation of mast cells observed in the perinatal period, initially without signs of systemic involvement. The typical driver mutation for cutaneous mastocytosis is in the *KIT* gene [2,3]. However, few reported cases have identified a molecular variant in the *KIT* gene [4,5,6,7]. The prognosis is poorly understood, as the disease has significant variability [1,2,3]. Thus, we aimed to present a case with NDCM and perform a literature review to provide a comprehensive overview of this rare pathology to better understand the prognosis.

## 2. Case Report

A male infant, without any prenatal evaluations, was born at term (40 weeks of gestation), birth weight of 3800 g, length at birth of 51 cm, and an APGAR score of 10. The mother had been smoking daily during pregnancy. His family history was unremarkable; he had healthy parents, a sister with vitiligo, and a brother with asthma. He is the mother’s fifth child. The patient and his family lived in a rural environment. The educational level of the mother was low, and she could not read or write, while the father had only completed primary school.

### 2.1. At Birth

He presented generalized subcutaneous nodules, on the face, scalp, trunk, back, hands, and feet. The palms and soles were spared of nodules (Figure 1A–C). At the age of 1 month, he showed a persistence of the cutaneous nodules, while the spleen and liver were normal in size. The Darier sign was positive, suggesting mast cell involvement. Clinical, imagistic (ultrasound), and laboratory work-up, including bone marrow biopsy and skin punch biopsy at age 2 months, were performed to understand the etiology of this unusual presentation.

### 2.2. At 2 Months of Age

The patient weighed 4.9 kg, with a body length of 53 centimetres (Figure 2). The abdominal ultrasound showed no signs of visceral malformations, or organomegaly. The ECG examination revealed a minor right bundle branch block. The blood count was in the normal range. Inflammatory markers had normal levels. He had elevated IgE levels, of 29.5 (normal below 15 UI/mL). Tryptase was significantly elevated at 540 μg/L (normal < 11 μg/L). The bone marrow biopsy showed no mast cells. Cardiac echocardiography showed a medium-sized atrial septal defect (ASD) ostium secundum type of 6.7 mm, a left-to-right shunt, right-sided dilatation of the heart chambers, and stenosis of the pulmonary arteries. Pulmonary hypertension was also detected. Thus, a diagnosis of a congenital heart defect was established. The bone marrow aspiration cytology interpretation and flow cytometry evaluation demonstrated hypercellular bone marrow with no mastocyte involvement at age 2 months. A punch biopsy revealed CD117-positive cells diffusely infiltrating the skin, and thus the diagnosis of NDCM was raised (Figure 1D–K). Recommendations for oral and local sodium cromoglycate and oral antihistamines were made. Treatment started 1 month later due to difficulties in access to the drug. The patient’s mother was advised that the baby would need urgent medical attention in case of skin redness, diarrhoea and/or vomiting with acute onset, or general malaise as they could represent a sign of systemic degranulation by the mastocytes. Additionally, a list of possible triggers for mastocyte degranulation was provided to the mother.

Subsequent DNA analysis, using Illumina Whole Exome Next Generation Sequencing, was performed from the FFPE–punch skin biopsy. Genomic DNA was quantified using the Qubit 4.0 fluorimetric assay (Thermo Fisher Scientific, Waltham, MA, USA) and sample integrity, based on the DIN (DNA integrity number), was assessed using a Genomic DNA ScreenTape assay on TapeStation 4200 (Agilent Technologies, Santa Clara, CA, USA). Libraries were prepared from 100 ng of total DNA using the NEGEDIA Exome clinical grade sequencing service (Next Generation Diagnostic srl, Pozzuoli, Napoli, 80078 Italy), which included library preparation, target enrichment using the Agilent V7 probe set, and quality assessment. Sequencing was completed on a NovaSeq 6000 sequencing system using a paired-end, 2 × 150 cycle strategy (Illumina Inc., San Diego, CA, USA). The raw data were analysed using the Next Generation Diagnostics srl Whole Exome Sequencing pipeline (v1.0) which involved a cleaning step by UMI removal, quality filtering and trimming, alignment to the reference genome, the removal of duplicate reads, and variant calling [9,10,11,12]. Variants were annotated using the Ensembl Variant Effect Predictor (VEP) tool5 (v. 104) [13]. Variant filtering included variants with coverage > 50×, population frequency < 0.1% in gnomAD, impact on protein, various in silico prediction models, and biological databases (Clinvar, COSMIC v94, ONKOKB). The median coverage of the sample was 197×, with 98.3% of bases covered > 50×. DNA sequencing identified a pathogenic [14], heterozygous variant in the KIT gene NM_000222.3:c.1504_1509dup; p.(Ala502_Tyr503dup), previously associated with cutaneous mastocytosis [2]. Therefore, the diagnosis of NDCM was confirmed.

### 2.3. At Age 4 Months

The patient weighed 5.7 kg with a body length of 61 centimetres (Figure 2). His bloodwork showed iron deficiency and anaemia. Cardiac echocardiography showed a diminished ASD of 6.3 mm and additional mitral and tricuspid valve regurgitation. Treatment included iron and calcium supplementation, antihistaminic (cetirizine), and local application and oral disodium cromoglycate.

### 2.4. At Age 1 Year and 2 Months

The patient presented to the hospital with respiratory symptoms compatible with pneumonia. He weighed 8.5 kg, with a body length of 70 cm (Figure 2). Bacterial infection of the nodular lesions of the scalp and secondary occipital and cervical adenopathy were present. An abdominal ultrasound examination revealed an enlarged spleen (splenomegaly) of 7.9 cm (normal 6.13 ± 0.13 cm). Additionally, right iliac fossa adenopathy and right and left inguinal adenopathy were noted. He had elevated inflammatory markers (C-reactive protein 54 mg/L) and reactive leukocytosis. He had normal levels of IgA, IgG, and IgM. He received antibiotic treatment, with subsequent improvement.

### 2.5. At Age 1 Year and 8 Months

The patient presented to the hospital due to fever and ear pain. His weight was 9.5 kg, and his body length was 78 cm (Figure 2). Bilateral otitis externa and adenotonsillar hypertrophy were diagnosed. Inflammatory markers were mildly elevated (11.5 mg/L). The bone marrow biopsy could not be performed, due to sedation constraints during respiratory infection. A chest X-ray revealed bilateral perihilar nodular lung opacities with a tendency to confluence. An ultrasound of the soft tissues demonstrated bilateral cervical adenopathy of approximately 11 mm, relatively hypoechoic on both sides of the neck. An abdominal ultrasound showed a liver with a homogeneous echostructure and normal size. The spleen was enlarged, measuring 8.6 cm (normal 6.45 ± 0.17 cm). The neurological examination showed delayed motor development, as he could maintain orthostatism independently only for a few seconds and could take steps with bilateral support. He did not have cerebellar or extrapyramidal signs. The psychological examination showed a mild delay in cognitive acquisitions, as expressive language included three words with a sense of communication, while receptive language was appropriate.

### 2.6. At Age 2 Years

The mother came with the boy for an outpatient care consultation yet refused hospital admittance despite the significant medical worry for the later-cervical tumour of more than 10 cm in diameter. At his age, he was still breastfeeding alongside eating various solid foods. The nodes had diminished on the face, yet the scalp had hundreds of nodules, apparently causing pruritus, as there were multiple scratching lesions that were infected (Figure 1L–N). At the time of that consultation, the patient was not receiving cromoglycate treatment. He was subsequently lost to follow-up.

## 3. Literature Review

A case aggregation of NDCM reported in the literature was performed to provide a comprehensive overview of this rare pathology [6,7,15,16,17,18,19,20,21,22,23,24,25,26,27,28,29,30,31,32,33,34,35]. We conducted a comprehensive literature review focusing on NDCM to provide an in-depth understanding of this rare pathology. Our primary source of data was Medline-indexed studies accessed via PubMed. The aim was to include articles that reported cases of NDCM, and cases of neonatal mastocytosis, especially those that highlighted clinical presentations, laboratory investigations, genetic and histopathological findings, and the outcomes of the disease. Our search included terms such as ‘neonatal mastocytosis’, ‘infant mastocytosis’, ‘cutaneous mastocytosis’ + ‘infant and/or neonatal’, ‘Darier sign’, ‘c-kit’, ‘*KIT* gene mutations’, ‘CD117-positive cells’, ‘skin biopsy’, ‘bone marrow biopsy’, and ‘molecular genetic testing’.

The literature review revealed a total of 31 cases of NDCM, including the present case, reported from 1957 to 2020, which are shown in summary in Table 1 and with details in Appendix A [6,7,15,16,17,18,19,20,21,22,23,24,25,26,27,28,29,30,31,32,33,34,35]. The male-to-female ratio of these cases was 20:11. The most common symptom was diffuse erythema and multiple nodules on the skin. Almost half of the cases had systemic involvement (defined by the involvement of another organ/system other than skin). The laboratory work-up for the diagnosis was complex. Elevated blood and/or urine markers (serum tryptase, serum histamine, and urine N-methylhistamine) were observed in half of the cases. The bone-marrow biopsy showed involvement in half of the cases where it was performed (7/31). A skin biopsy was performed in most cases (27/31), yet special stains (Toluidine Blue and Giemsa were most used) were used only in 12 cases, while immunohistochemistry (IHC) was performed in 8 cases. In the literature, variants in the *KIT* gene reported in NDCM cases were identified in only four other cases, as follows: three cases with p.(Asp816Val), and dup A502Y503 in one case [4,5,6,7]. Some reported cases of NDCM had negative molecular genetic testing results (five cases), while for the majority genetic testing was not performed (Table 1). The main treatment involved antihistamines and/or cromoglycate in most cases (23/31), with systemic corticoids for systemic manifestations (8 cases), 1 case with Vitamin K and 1 case with iron preparation. The longest follow-up reported was 27 years, and the vital status at follow-up was mostly alive. However, data extracted from literature reports could not accurately show disease progression.

## 4. Discussion

NDCM is a rare form of cutaneous mastocytosis that presents a wide range of clinical manifestations, diagnostic findings, and treatment approaches. This discussion aims to compare the findings from a recent case with those from a comprehensive review of 31 cases in the literature.

The recent case shows a male patient, consistent with the male predominance observed in the literature [4,5,6,7,14,17,18,19,20,23,24,27,29,30,31]. The positive Darier sign in the present case aligns with most of the reviewed cases [5,6,7,15,16,17,18,21,22,24,26,29,30,31,32,33], indicating the commonality of this feature in NDCM. Clinical presentation and the severity of disease showed significant variability between the 31 reviewed cases. Systemic involvement was absent in the present case in the first year, yet developed after the age of 1 year, consistent with the literature where systemic involvement was reported in cases by Shah et al. [4], Heide et al. [5], Waters et al. [18], Allison et al. [20], Burgoon et al. [21], and Harrison et al. [23]. However, cases with and without systemic involvement were reported, showing variability in the presentation of NDCM.

Diagnostic markers (serum tryptase, serum and urine histamine, urine N-methylhistamine) in the present case were elevated blood tryptase, similar to cases reported by Heide et al. [5], Chaudhary et al. [6], Jenkinson et al. [7], Mann et al. [27], Walker et al. [28], Duckworth et al. [29], Koga et al. [30], Lange et al. [31], Ghiasi et al. [32], Folch et al. [34], and Turnbull et al. [35] that reported elevated markers. However, urine N-methylhistamine was not performed in this case. The bone marrow biopsy in the present case did not show involvement, aligning with most of the reviewed cases [4,6,21,28,32].

The special stains used in the histologic evaluation of biopsies, particularly Toluidine Blue in the present case, were consistent with the literature [6,15,16,21,22,24,28,30,32,35]. However, ultrastructural analysis was not conducted in the present case, similar to most of the reviewed cases, as this type of analysis is largely inaccessible and not necessary for diagnosis [15,16]. IHC in the present case showed diffuse positivity for CD117 (c-KIT), a finding that aligns with four cases in the literature [31,33,35]. The identification of the *KIT*—c.1504_1509dup p.(Ala502_Tyr503dup), to our knowledge, was not reported before in the literature [6,7,15,16,17,18,19,20,21,22,23,24,25,26,27,28,29,30,31,32,33,34,35]. A small number (4 out of 31) of cases in the literature reported a *KIT* pathogenic variant. Specifically, the reported *KIT* variants were p.(Asp8l6Val) in three cases and *KIT* dup A502Y503 in one case [4,5,6,7]. Some authors have only tested the p.(Asp8l6Val) variant, without *KIT* gene sequencing, and therefore this approach could explain the negative results. In a study by Bibi et al. 2014, 75% of paediatric patients had variants in *KIT*, supporting the assumption that childhood mastocytosis is a clonal disease [1]. The authors noted that the spectrum of the disease and variants are limited compared to those seen in adults [1]. The genetic foundations of mastocytosis are closely tied to mutations in the c-kit gene, with particular significance attributed to exon 18. Wöhrl et al. [8] highlighted a mutation in this exon in the context of familial mastocytosis [8,36]. Their findings emphasize the pivotal role exon 18 mutations play in the pathogenesis of this disease and underscore the importance of comprehensive genetic screening in patients with suspected mastocytosis. With a further characterization of molecular causes in NDCM, future genotype-phenotype correlations could be proposed. The treatment in the present case included Cromoglycate and antihistamines, aligning with most of the cases in the literature [4,5,6,17,26,30,31,33,34,35]. Although Cromoglycate is the mainstream medication for managing the symptoms of NDCM, the treatment efficiency in preventing complications is limited in some cases. Further research is needed to provide improved therapeutic approaches. The present case developed complications, including splenomegaly and cervical adenopathy, after 2 years of follow-up. This aligns with the literature, where complications were reported in cases by Shah et al. [4], Heide et al. [5], Chaudhary et al. [6], Willemze et al. [15], Sethuraman et al. [17], Waters et al. [18], Yasuda et al. [19], Allison et al. [20], Burgoon et al. [21], Klaber et al. [22], Fernandes et al. [26], Lange et al. [31], Ghiasi et al. [32], and Folch et al. [34]. In terms of follow-up duration, the present case had a follow-up of 2 years, which is within the range observed in the literature. The follow-up durations in the literature varied widely, from as short as 8 days to as long as 27 years [15]. This wide range underscores the chronic nature of NDCM and the need for long-term monitoring and management.

## 5. Conclusions

The present case provides additional insights into the variability of clinical presentation, diagnosis, and management of this precocious and severe form of neonatal diffuse cutaneous mastocytosis. The variant c.1504_1509dup in the *KIT* gene has not been reported before, in association with this form of disease. The case aggregation of reported affected individuals in the literature provides a comprehensive overview of this rare pathology. Most of the affected individuals had severe complications, such as mast hepatosplenomegaly, adenopathy, bacterial infections, and/or mast cell leukaemia. Systemic involvement was observed in almost half of the cases. A gene-to-phenotype correlation cannot be made with this limited number of cases. Several cases presented did not have genetic testing performed at all, or limited genetic testing was performed only for the p.(Asp8l6Val) variant. With the further characterization of molecular causes in diffuse cutaneous mastocytosis, future genotype-phenotype correlations could be proposed. Additionally, the skewed male-to-female ratio (2:1) raises the possibility that sex differences could play a role in pathogenesis. Despite existing treatment with Cromoglycate, more effective therapeutic strategies are needed.

## Figures and Tables

**Figure 1 genes-14-02021-f001:**
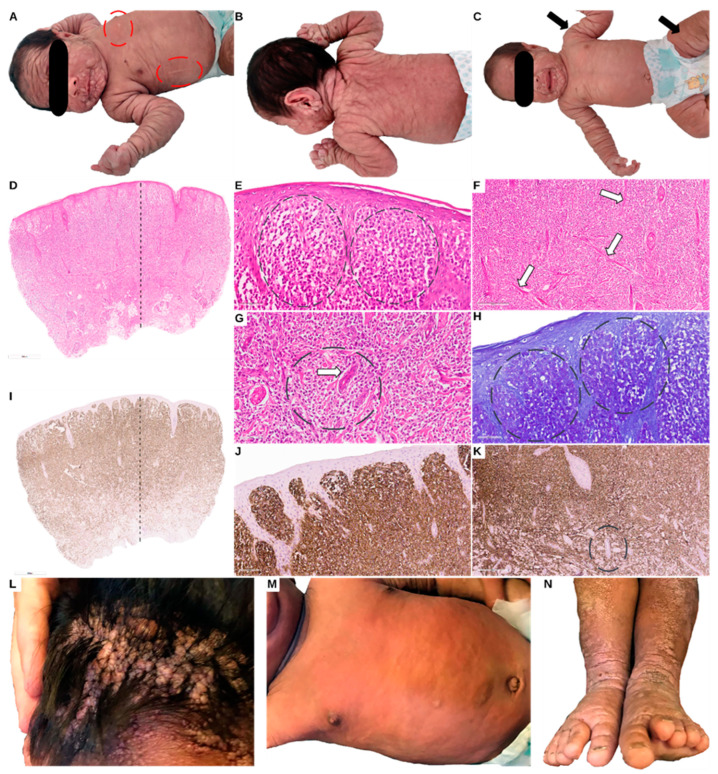
Clinical presentation at age 1 month with generalized subcutaneous nodules, including on the face, scalp, hands, and feet (Panels (**A**–**C**)). Darier’s sign was positive, suggesting mast cell degranulation (ovals Panel (**A**)). Punch biopsy of two skin nodules (Panel (**C**) arrows indicating the site of the punch biopsy) showed a thin epidermis and the diffuse infiltration of mast cells in the superficial dermis with extension in the deep dermis, some forming aggregates around adnexal structures and blood vessels. Panel (**E**) shows aggregates of mast cells in the superficial dermis, panel (**F**) arrows show bundles of collagen, and in panel (**G**) the oval and arrow indicate a perivascular aggregation of mast cells (Panels (**D**–**K**)). Toluidine Blue shows mast cells with intracytoplasmic purple granules (Panel (**H**)). The mast cells were diffusely positive for CD117 (Panels (**I**–**K**)). Panels (**L**–**N**): clinical presentation at the age of 2 years. Panel (**L**) shows aggregates of nodules on the scalp, panel (**M**) shows the thorax and abdomen, with less visible signs of nodules, and panel (**N**) shows legs and feet with visible plaque-like lesions showing hyperkeratosis.

**Figure 2 genes-14-02021-f002:**
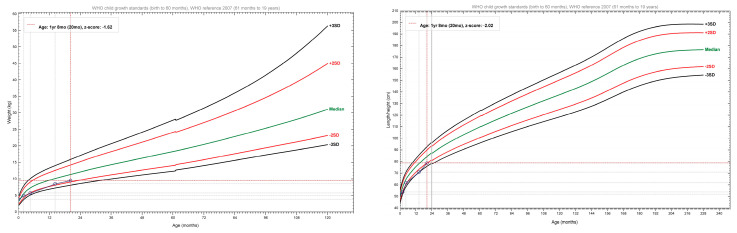
Plotted growth parameters of the patient, using the World Health Organization reference [8]. WHO. Anthro App Software for Assessing Growth of the World’s Children and Adolescents; Available online: https://www.who.int/tools/growth-reference-data-for-5to19-years (accessed on 5 May 2020).

**Table 1 genes-14-02021-t001:** Literature review: summary of 31 cases of NDCM reported in the literature reported from 1957 to 2020. NA—not available.

	Present Case	Case Aggregation of DCM from the Literature
Number of cases	1	31
Sex (Male/Female)	M	19 M, 11 F, 1 NA
Darier sign (positive, negative or no information)	Positive	20 positive, 1 negative, 10 and NA
Systemic involvement (SI)	Yes	9 had SI, 13 without SI, and 9 NA
Positive (elevated) blood and/or urine markers	Yes, elevated serum tryptase	15 had elevated markers, 7 had normal markers, and 9 NA
Bone-marrow biopsy (BMB)	Yes—no involvement	7 had BMB, 13 without BMB, and 11 NA
		Out of the 7 that had BMB, 4 had BM involvement
Biopsy	Yes—skin	27 had a biopsy performed (with skin being the most common), 2 without biopsy, 2 NA
Special stains	Toluidine Blue	12 cases had special stains, (Toluidine Blue and Giemsa most used), 8 had no special stains, 11 NA
Ultrastructural analysis (UA)	No UA	2 had UA, 26 without UA, and 3 NA
IHC	CD117 (c-kit)—diffuse positivity	5 had IHC for CD117, 3 had IHC for Tryptase, 9 no IHC performed, and 14 NA
Variant testing	*KIT*—c.1504_1509dup p.(Ala502_Tyr503dup)	3 cases with *KIT* p.(Asp8l6Val), *KIT* dup A502Y503
Treatment	Cromoglycate	The main treatment was antihistamines and/or cromoglycate (23 cases) with systemic corticoids for systemic manifestations (8 cases), 1 case with Vitamin K and 1 case with iron preparation, and 6 cases with NA
Follow-up	2 years	The longest follow-up was 27 years
Complications	Splenomegaly and cervical adenopathy	Complications were reported in 17 cases with the most common complication being spleen and/or hepatomegaly, 10 had none reported, 4 NA
Vital status at follow-up (alive or dead)	Alive	24 were alive at follow-up, 6 were dead with 1 death of unknown cause

## Data Availability

The data that support the findings of this study are available on request from the corresponding authors. The data are not publicly available due to privacy or ethical restrictions.

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
