# Peer review of "When Mast Cells Run Amok: A Comprehensive Review and Case Study on Severe Neonatal Diffuse Cutaneous Mastocytosis"

_genes, 2023, doi:10.3390/genes14112021_

Round 1

Reviewer 1 Report

Comments and Suggestions for Authors

The article is fascinating and well-written and shows an infrequent and little-known condition related to a new mutation in the KIT gene in the literature.

The images in the panel are very well taken and organised, also well described in the caption.

I also appreciated the detailed description of the follow-up and diagnostic work-up.

The only subtlety that I would like to highlight is the definition of the condition as “neonatal” rather than congenital in light of the findings at birth of the clinical manifestation and the presence of a linked genetic mutation.

Furthermore, I would like to see the summary table a little better organized, as I found a little bit confusional.

Finally, I would ask you to include in the manuscript the databases in which the searches were performed and the search strings used.

Author Response

Dear reviewer, 

We would like to thank you for the encouraging and appreciative words regarding our work, we appreciate it. Thank you!

We have tweaked the summary table so it is more clear and easy to understand. 

We have added the strategy for the literature review and the search strings used in the literature search. 

Regarding the term congenital vs neonatal, we have chosen to maintain the word neonatal even if congenital would more properly describe a condition that is already present at birth. The factual reason we have chosen to stay with the term neonatal is that this term has permeated the scientific literature in the description of these cases. 

All changes have been highlighted for better tracking.

Once again we thank the reviewer for the encouraging review and for the time invested in the betterment of our manuscript. 

The authors. 

Reviewer 2 Report

Comments and Suggestions for Authors

When Mast Cells Run Amok: A Comprehensive Review and 2 Case Study on Severe Neonatal Diffuse Cutaneous Mastocytosis .

Emilian-Gheorghe Olteanu and co-authors have prepared an interesting paper, combining data on a clinical case of neonatal diffuse cutaneous mastocytosis with the literature data available to date. Thus, the collection of scientific data on this rare pathology was replenished and measures were proposed to improve treatment protocols for such patients and clarify the prognosis of the disease.

The published data from a clinical case of a rare disease certainly deserve attention. However, despite the importance of the work, for a more complete understanding of the pathogenesis of the disease, introducing novelty into the pathogenetic significance of MC, and clarifying the diagnosis, the reviewers would recommend that the authors try to conduct more original research on the state of the MC population in this pathology. Relying on toluidine blue or CD117 staining alone to detect mast cells does not mean that you will see the entire mast cell population (see (Atiakshin et al. 2017)).

In addition, with this methodological approach, the authors are deprived of the opportunity to evaluate the secretory activity of MC and what kind of mediators the effect of MC on the structural components of the skin occurs. It is more objective in this case to use the determination of the profile of specific MC proteases with the detection of tryptase, chymase and carboxypeptidase A3. This method will allow to discover new pathogenetic features of the development of the disease, the interaction of MCs with other representatives of the immune landscape, and, possibly, offer new ways of personalized therapy.

Atiakshin D, Samoilova V, Buchwalow I, Boecker W, Tiemann M (2017) Characterization of mast cell populations using different methods for their identification. Histochem Cell Biol 147 (6):683-694. doi:10.1007/s00418-017-1547-7

Author Response

Dear reviewer,

We thank you for taking the time to review our paper titled "When Mast Cells Run Amok: A Comprehensive Review and Case Study on Severe Neonatal Diffuse Cutaneous Mastocytosis" and appreciate your constructive comments.

Regarding the methodology, we would like to clarify a few points:

Scope of Our Study:

It's worth noting that the primary scope of our study was not to determine the profile of specific mast cell (MC) proteases. While we acknowledge the added depth and specificity such an analysis would offer, our intention was to focus on the broader patterns and implications related to neonatal diffuse cutaneous mastocytosis.

Choice of Toluidine Blue and CD117 Staining:

Our choice of using toluidine blue and CD117 as staining markers was deliberate. We based this choice on the vast majority of studies in the literature that utilize these staining markers for similar investigations. Furthermore, we used CD117 specifically because of evidence in the literature indicating that a presumed c-kit mutation may lead to CD117 overexpression. As highlighted in our paper, this was consistent with our findings, where we noted the infiltration of diffusely strong positive CD117 cells in the superficial dermis and dermis.

We recognize the value of the methodological approach you suggested, particularly in uncovering new pathogenetic features of the disease and understanding the interactions of MCs with other representatives of the immune landscape. However, the scale and focus of our current research were more aligned with providing a comprehensive overview and correlating our findings with existing literature, using widely accepted staining markers.

Nevertheless, we are grateful for your feedback and will consider the inclusion of more specific MC proteases and the determination of tryptase, chymase, and carboxypeptidase A3 in future research endeavours to uncover deeper insights into the pathogenesis of the disease.

We thank you once again for your thoughtful review and recommendations.

Sincerely,

Emilian-Gheorghe Olteanu and Co-authors

Reviewer 3 Report

Comments and Suggestions for Authors

Mastocytosis are classified into diffuse cutaneous mastocytosis (DCM), urticaria pigmentosa (UP) and solitary mastocytoma. A case of DCM, appearing at the birth (NDCM) is presented.

The work is interesting, complete, explains the diagnostic investigations, the evolution and the treatment.

To these forms of mastocytosis I would like to mention familial mastocytosis (FM), in which some people in the family are affected, for example brothers/sisters (Trevisan G, Pauluzzi P, Gatti A, Semeraro A. Familial mastocytosis associated with neurosensory deafness. JEADV 2000, 14: 119-122).

I would also add a comment on the genetics of mastocytosis and in particular on the c-kit mutation in exon 18 (Wöhrl S, Moritz KB, Bracher A, Fischer G, Stingl G, Loewe R. A c-kit mutation in exon 18 in familial mastocytosis. J Invest Dermatol. 2013 Mar;133(3):839-841. doi: 10.1038/jid.2012.394. Epub 2012 Nov 29).

The work requires minor revision and can therefore be accepted.

Author Response

Dear Reviewer,

Thank you for the supportive and constructive evaluation of our work. 

We have now added all the mentioned works in the revised manuscript and we have highlighted the changes for better tracking.

Thank you again for the time invested in the betterment of our manuscript.

The authors.